# The Loop Tenodesis Procedure—From Biomechanics to First Clinical Results

**DOI:** 10.3390/jcm10030432

**Published:** 2021-01-23

**Authors:** Moritz Riedl, Agnes Mayr, Stefan Greiner, Christian Pfeifer, Isabella Weiss, Lina Forchhammer, Volker Alt, Maximilian Kerschbaum

**Affiliations:** 1Clinic of Trauma Surgery, University Medical Center Regensburg, Franz-Josef-Strauss-Allee 11, 93053 Regensburg, Germany; moritz.riedl@ukr.de (M.R.); agnes-mayr@t-online.de (A.M.); christian.pfeifer@ukr.de (C.P.); isabella.weiss@stud.uni-regensburg.de (I.W.); lina.forchhammer@stud.uni-regenburg.de (L.F.); volker.alt@ukr.de (V.A.); 2Sporthopaedicum Straubing, Bahnhofplatz 27, 94315 Straubing, Germany; greiner@sporthopaedicum.de

**Keywords:** shoulder, arthroscopy, long biceps tendon, tenotomy, tenodesis, LHB

## Abstract

(1) Introduction: Several surgical therapy options for the treatment of pathologies of the long biceps tendon (LHB) have been established. However, tenotomy, as well as established tenodesis techniques, has disadvantages, such as cosmetic deformities, functional impairments and residual shoulder pain. This study presents the first clinical and structural results of the recently introduced loop tenodesis procedure for the LHB, developed to overcome these issues. (2) Methods: 37 patients (11 women, 26 men, mean age 52 years), who underwent loop tenodesis of the LHB were examined six months after surgery. For the clinical evaluation the Constant score, as well as the LHB score, were used, complemented by elbow flexion and supination strength measurements. The integrity of the tenodesis construct was evaluated indirectly by sonographic detection of the LHB in the bicipital groove. (3) Results: Both, the overall Constant score as well as the LHB score showed significant improvements six months postoperatively, as compared to the preoperative value. Fourteen patients (38%) presented an examiner-dependent upper arm deformity, although only five patients (13%) reported subjective cosmetic deformities. Both, flexion and supination strength were preserved compared to the preoperative level. In 35 patients (95%), the tenodesis in the bicipital groove was proofed sonographically. (4) Conclusion: The loop tenodesis of the LHB provides good-to-excellent overall clinical results after a short-term follow-up of six month. The incidence of cosmetic deformities was inferior compared to conventional therapy options (tenotomy and anchor tenodesis).

## 1. Introduction

Pathologies of the long head of biceps tendon (LHB) are a common cause of anterior shoulder pain [1,2]. Several open and arthroscopic surgical therapy options for treatment of the long biceps tendon have been established. The simple tenotomy is confronted with various tenodesis techniques [3,4]. The tenotomy is a minimally invasive and easily performable arthroscopic procedure; however, it has a high risk of distalisation of the tendon and might cause muscle cramps and cosmetic deformities of the upper arm [5,6]. Thus, a tenodesis of the LHB with strong tendon-to-bone fixation by anchor or interference screw is recommended to reduce distalization and subsequent complications, as shown by multiple studies [7,8,9]. Nevertheless, some implant-associated disadvantages of these procedures are reported. Besides fractures, implant dislocation and nerve injuries, as well as residual anterior shoulder pain at the insertion site are reported complications [7,10,11,12]. The so-called loop tenodesis was developed to address these issues in the treatment of long biceps tendon pathologies [13]. Based on the principle of “autotenodesis” the technique supports the tendon’s self-locking mechanism in the bicipital groove by creating a tendon loop at the tenotomized proximal LHB and enlarging its diameter to prevent the tendon from distalization. In a previous study we investigated the biomechanical properties of the tenodesis construct and the results revealed a significantly higher stability of the loop tenodesis compared to a simple tenotomy of the LHB [14]. In the biomechanical setting, the loop tenodesis tolerated significantly higher loads and showed higher stiffness with less distalization. Failure of the loop tenodesis required a complete rupture distal of the loop, while the tenotomy failed because of a slippage of the tendon through the bicipital groove. Thus, the loop tenodesis procedure combines the advantages of both a simple tenotomy and a biceps tenodesis by fixing the tendon stably without any implant in a fast and minimally invasive manner.

The aim of this work was to investigate the clinical, cosmetic and structural outcome of the novel loop tenodesis procedure for the long head of biceps.

## 2. Materials and Methods

This prospective clinical trial included a cohort of 55 patients with degenerative rotator cuff lesions undergoing the loop tenodesis procedure of the LHB between May 2018 and November 2018. Indications for LHB tenodesis were partial or subtotal tears of the LHB, chronic LHB inflammation, LHB instability due to pulley lesions and degenerative SLAP (superior labrum from anterior to posterior) lesions. The exclusion criteria of the study were osteoarthritis, shoulder stiffness, shoulder instability, distal biceps tendon lesions or previous operations on the contralateral shoulder. All patients received detailed information about study objectives, surgical techniques, as well as examination methods. The patients were evaluated in preparation for the intervention as well as six weeks and six months after surgery. Informed consent was obtained. This study has been approved by the local ethical committee (18-1032-101). All patients have been treated by the same shoulder surgeon. Table 1 list the demographic data of the patient cohort included in the six month follow up.

### 2.1. Surgical Technique

According to the previously published technique of the loop tenodesis procedure [13], the LHB was cut close to its origin at the superior glenoid labrum using an electrothermal instrument (Figure 1A), retrieved by a clamp through an antero-lateral portal and sutured in a loop configuration after resection of the proximal 0.5–1 cm of the LHB. After a loop was created and fixed by suture (Figure 1B), the tendon was released and shuttled back intraarticularly. The tendon loop locks itself stably at the cranial entrance of the bicipital groove, which was evaluated arthroscopically (camera and setup: Arthrex, Naples, FL, United States; optics: Karl Storz SE and Co. KG, Tuttlingen, Germany) (Figure 1C). Any necessary procedures for the treatment of concomitant pathologies are performed after finalization of the loop tenodesis procedure. Figure 1 illustrates the single steps of the loop tenodesis procedure.

### 2.2. Postoperative Rehabilitation and Treatment

According to the recommendations for arthroscopic tenodesis procedures, all patients undergo a supervised physical therapy program. To protect the LHB, the elbow is fixed in 90° flexion and in a neutral rotation position for 4 weeks using a sling. The protocol includes passive movement from day 2 postoperatively until 6 weeks after surgery. Patients are informed to avoid any elbow flexion and supination maneuvers against resistance for 6 weeks, including no weight bearing at the operated upper extremity. Further, an upper-arm bandage has to be worn for 6 weeks after surgery to promote stable healing of the LHB loop in the initial phase and prevent distalization of the muscle. Depending on the presence of additionally performed procedures, alterations of the postoperative rehabilitation protocol can be required. For instance, based on the severity of the rotator cuff lesion, the patient was prescribed a shoulder abduction cushion/immobilizer between 4 and 6 weeks.

### 2.3. Functional Evaluation

For assessment of global shoulder function, the Constant score was evaluated [15]. The Constant score (maximum 100 points) includes 4 subsections: ‘‘pain’’ (maximum 15 points), ‘‘activities of daily living’’ (maximum 20 points), ‘‘active range of motion’’ (maximum 40 points), and ‘‘strength’’ (maximum 25 points). In order to determine the function of the long biceps tendon precisely, the LHB score and supination strength were examined in each patient [9,16]. The LHB score (maximum 100 points) assesses the three qualities ‘‘biceps pain and muscle cramps’’ (maximum 50 points), ‘‘cosmesis’’ (maximum 30 points) and ‘‘flexion strength at the elbow’’ (maximum 20 points). Measurement of elbow flexion strength was performed in 90° of elbow flexion using an isometric dynamometer (IsoBex Dynamometer, MDS AG, Burgdorf, Switzerland) and repeated three times. Supination strength was measured in 90° of elbow flexion and neutral forearm rotation with a Baseline hydraulic dynamometer (Fabrication Enterprises Inc., White Plains, NY, USA). The measurement of both elbow flexion and supination strength was performed on the affected side, as well as on the contralateral side by an independent examiner.

### 2.4. Structural Evaluation

In all patients, a standardized ultrasound evaluation of the operated shoulder was performed using a multifrequency (15–6 MHz) linear array transducer and a Aloka Prosound 6 ultrasound system (Hitachi, Ltd., Tokyo, Japan). The location of the LHB in the bicipital groove was examined in transversal and longitudinal ultrasound plains. Six months after surgery, detection of the LHB in the bicipital groove was assessed as (auto-)tenodesis, whereas absence of the LHB was defined as structural failure of the loop tenodesis technique.

### 2.5. Statistical Evaluation

The statistical analysis using repeated measures ANOVA, Friedman test as well as Bonferroni post-hoc test (level of significance, *p* = 0.05) was carried out using SPSS software (SPSS Inc, Chicago, IL, USA).

## 3. Results

Fifty-five patients (16 ♀, 39 ♂; ø 53 years) were examined preoperatively and six weeks after surgery. Thirty-seven patients (11 ♀, 26 ♂; ø 52 years) were included in the six months follow-up. Eighteen patients were not reachable for the latest follow-up investigation. The original loop tenodesis technique was performed on all patients. The dominant side was affected in 24 patients, whereas surgery on the non-dominant side was performed on 13 patients (Table 1).

### 3.1. Intraoperative Findings and Concomitant Procedures

The most frequent LHB pathology was a partial LHB tear, which was found in all 37 cases. Pulley lesions were detected arthroscopically in 35 patients. Furthermore, 12 patients exhibited LHB tendinitis. A SLAP II lesion was determined in one case. The intraoperative findings were identified macroscopically.

In addition to the loop tenodesis, all patients received a rotator cuff repair as well as an arthroscopic subacromial decompression. In one patient an arthroscopic resection of the acromioclavicular joint was performed (Table 2).

### 3.2. Functional Results

Clinical results six months after surgery showed a mean Constant score of 80 ± 13 points (range 31–98 points) on the operated side compared to 62 ± 17 points (range 26–100 points) preoperatively and 50 ± 21 points (range 23–97 points) six weeks postoperatively. Taking the subcategories into account, the average score for “pain” reached 11 ± 3 points (range 0–15 points), for “activities of daily living” 17 ± 3 points (range 5–20 points), for “active range of motion” 37 ± 4 points (range 22–40 points) and for “strength” 15 ± 7 points (range 4–25 points) six months after surgery. The differences between the preoperative and 6 months postoperative examination reaches statistical significance (*p <* 0.05) regarding the overall constant score as well as the subcategories “pain”, “activities of daily living”, “active range of motion”. The measurement of the strength showed no significant difference between preoperative strength compared to strength six months postoperatively (Figure 2).

For more accurate assessment of long biceps tendon, the LHB score was evaluated in all patients. A mean LHB Score of 85 ± 11 points (range 56–100 points) was determined six months after surgery in comparison with 75 ± 13 points (range 42–93 points) preoperatively and 77 ± 11 points six weeks postoperatively. The subcategory “biceps pain and muscle cramps” reached an average of 44 ± 6 points (range 31–50 points), “cosmesis” 26 ± 7 points (range 0–30 points) and “flexion strength at the elbow” averaged 14 ± 6 points (range 0–20 points). A significant difference (*p <* 0.05) between preoperative values and six months postoperatively was detected for the total LHB score and the subcategories “biceps pain and muscle cramps” and “cosmesis” (Figure 3). A significant difference in “strength at the elbow” of the preoperative examination compared to 6 months after surgery did not exist.

Regarding the entity cosmesis of the LHB score, the mean cosmetic result is evaluated patient-dependent as well as examiner-dependent. Six months after loop tenodesis, a significant difference (*p <* 0.05) between patient-dependent and examiner-dependent evaluation of the cosmetic results was determined. Two patients complained about a mild upper arm deformity as well as two patients stated a moderate cosmetic deformity. One patient displayed a severe Popeye deformity 6 months after surgery. However, the examiner detected a mild upper arm deformity in 11 patients, a moderate deformity in two patients, as well as a severe Popeye deformity in 1 patient (Table 3).

In order to evaluate the functional outcome of the loop tenodesis technique, supination strength was measured at every examination. Figure 4 shows the supination strength of the operated side in the percentage of the non-operated side. Six months postoperatively, the mean supination strength reached 82% compared to 72% preoperatively and 47% six weeks postoperatively. A significant difference (*p <* 0.05) of the preoperative supination strength compared to the examination six months postoperative could not be observed.

### 3.3. Structural Evaluation

In the ultrasound assessment six months postoperatively, (auto-)tenodesis of the LHB was detected in 35 patients. In two patients, a complete absence of the LHB loop was observed sonographically in the bicipital groove. One of these two patients developed severe Popeye deformity, the other patient showed moderate cosmetic deformity.

## 4. Discussion

The therapy options for pathologies of the LHB have been compared in numerous studies. Biceps tenodesis, as well as simple tenotomy, achieve good functional results [7,17,18,19,20]. However, biceps tenodesis holds advantages concerning cosmetic results and elbow flexion and supination strength and is recommended to avoid complications resulting from a distalization of the LHB such as muscle cramps, upper arm deformities, or functional impairment [6,7,19]. Nevertheless, some implant-associated disadvantages of these procedures are reported, including fractures, implant dislocation, nerve injuries and residual anterior shoulder pain [7,10,11,12]. In contrast, a complete slippage of the tendon through the bicipital groove is a very rare consequence of the LHB tenotomy [5].

The recently introduced loop tenodesis procedure [13] is based on the self-locking potential of the tenotomized LHB at the bicipital groove [5]. Enlarging the proximal diameter of the tendon by forming a loop after tenotomy allows for a stable autotenodesis of the LHB at the entrance to the bicipital groove [13]. This technique combines the advantages of both, simple tenotomy and tenodesis. This technique is quickly feasible, just like a tenotomy, and leads to a stable fixed tendon without an anchor or other fixation implant.

This article provides an overview of the procedure and outlines the first clinical and structural results six months after LHB treatment using the innovative loop tenodesis technique.

Below, the collected data are compared to a historic collective of patients treated with tenotomy and arthroscopic knotless suprapectoral tenodesis, which were evaluated with the same methodology used in the present study [5,21]. Therefore, a cohort of patients with degenerative rotator cuff lesions and the exact same examination methods were chosen to provide similar conditions.

The surgical technique was performed in all patients as a standard procedure, as explained above. The adjustment of the technique to pathologies of the long biceps tendon, such as LHB tendinitis, as well as to concomitant procedures, e.g., in massive rotator cuff lesions, will have to be studied and evaluated closely in the future [22,23].

The present study shows good to excellent overall clinical results for the novel loop tenodesis technique. The global shoulder function, represented by the Constant score, displayed significant improvement regarding the total score as well as the qualities “pain”, “activities of daily living” and “active range of motion” six months postoperative. A preservation of strength could be determined. These results are comparable to those of previous studies examining conventional surgical techniques [5,21]. However, for the assessment of LHB patients, the choice of the right evaluation system is of particular importance. Compared to other shoulder pathologies global shoulder function scoring systems, e.g., the Constant score, are not sensitive enough in these cases as their results are basically dependent on concomitant pathologies and their recovery (e.g., rotator cuff repair), potentially resulting in favorable score values in spite of present LHB pathologies [17,19]. For this reason, the LHB score was developed to more accurately evaluate LHB associated outcomes as it includes biceps related items such as muscle deformity, cramps and flexion strength [9,16].

The patients included in the study showed a statistically significant improvement in the overall LHB score and in the subcategory “biceps pain and muscle cramps” six months after loop tenodesis of the LHB and achieved comparably good results regarding previous studies. In particular, the significant improvement in the category “biceps pain and muscle cramps” leads to the assumption that the loop in the anterior aspect of the shoulder does not cause conflict and therefore does not lead to prolonged anterior shoulder pain, which is reported as a major implant-associated disadvantage of other tenodesis techniques [7,10,11,12].

In addition to the categories included in the LHB score, measurement of supination strength is an important tool for the comprehensive evaluation of the long biceps tendon [6,24]. The supination strength measurement six months postoperatively revealed a reduced supination strength of 82% compared to the non-operated side. Previous studies showed a comparable value of 85% after tenotomy and 106% after tenodesis. However, the mean follow-up time in these studies was 39 months compared to six months in the present study. A further improvement of strength after a longer follow-up might be expected.

Regarding elbow flexion, preservation of strength was demonstrated six months postoperatively compared with the baseline.

The overall cosmetic result of arthroscopically performed loop tenodesis was excellent in comparison to common surgical techniques. Whereas mild upper arm deformities occurred in 69% of patients after performing tenotomy and tenodesis, the present study detected cosmetic deformities in only 38% of patients [5]. Compared to tenotomy and anchor fixation the loop tenodesis is capable of achieving a more physiological length-tension relationship reducing the incidence of cosmetically relevant deformities. The low, but still remaining rate of deformities after loop tenodesis, might be due to secondary elongation of the LHB.

The subcategory “cosmesis” of the LHB score demonstrated a statistically significant difference between examiner and patient-dependent outcome. This observation is supported by the current literature. Walch et al. ascribes the discrepancy between subjective and objective impression to the reduced subjective perception of the muscle deformity, especially in older patients [25]. Similar results are shown by Scheibel et al. and Osbahr et al. [9,26]. It should be noted that some deformities developed between six weeks and six months postoperatively. We attribute this to the onset of the active range of motion and flexion and supination against resistance six weeks after surgery.

Within the study cohort, the sonographical proof of an autotenodesis of the LHB in the bicipital groove was lacking in only two patients. The low sonographical failure rate of 3.6% is in concordance with the absence of severe muscle deformities in most patients.

As the cosmetic results are based on subjective evaluation measurements of the arm circumference would feature an objective investigation tool for upper arm deformities and should be included in following studies. However, the used LHB score with its patient-dependent and examiner-dependent evaluation of the cosmetic outcome is considered to be a valid tool to assess the postoperative clinical progress as well [16].

Postoperative rehabilitation was performed according to our standard protocol for tenodesis with the restriction of forced elbow flexion and supination for six weeks. A recent study showed that a more conservative approach could contribute to limited ROM after tenodesis [27]. However, earlier active motion of the tenotomy group may have resulted in a higher number of Popeye deformities. Therefore, we opted for the more conservative rehabilitation protocol in analogy to conventional tenodesis techniques to ensure stable healing of the LHB loop.

Nevertheless, the following limitations of the present work must be considered. First, the study lacks a suitable control group. A randomized design comparing the loop tenodesis technique with the conventional techniques of tenotomy and tenodesis is required. Nevertheless, our historical patient collective (tenotomy, anchor tenodesis) is suitable for comparison as we used the identical study design for the present work. Secondly, this study presents the first clinical results six months after loop tenodesis of the LHB was performed. A longer follow-up is needed to draw more consistent conclusions.

## 5. Conclusions

The innovative implant-free loop tenodesis procedure for the long head of biceps shows favorable functional and cosmetic outcomes. Even after a short-term follow-up of six months, the outcome is already comparable to those of conventional therapy options, i.e., tenotomy and anchor tenodesis. However, long-term results have to follow to prove the sustainable success of this promising procedure.

### Outlook: All-Inside Loop Tenodesis Technique

To further reduce the invasiveness of the original loop tenodesis technique, an all-inside loop tenodesis technique was developed and published in 2019 by Kerschbaum et al. [28]. By the use of a special stitch configuration, the LHB loop can be created in an all-arthroscopic manner with only two arthroscopic portals and without extracorporeal treatment of the tendon. Compared to the arthroscopy-assisted technique [13], this approach further reduces the stress on the tendon and the potential risk of infection.

## Figures and Tables

**Figure 1 jcm-10-00432-f001:**
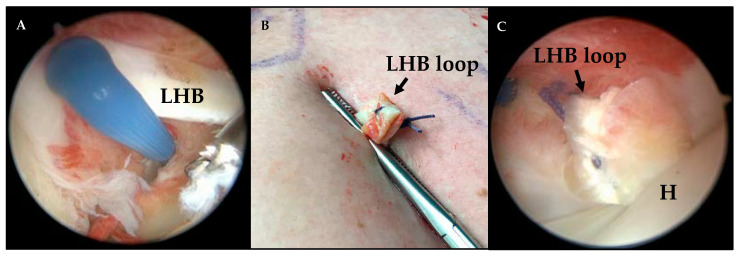
Right shoulder from the posterior portal with the patient placed in beach chair position. (**A**) Tenotomy of the biceps tendon close to its insertion; the biceps tendon is pulled extraarticularly through the anterolateral portal. (**B**) Long biceps tendon after extracorporeal creation of the loop and fixation with suture. (**C**) Long biceps tendon loop locked at the entrance of the bicipital groove (H, humeral head; LHB, long head of biceps.).

**Figure 2 jcm-10-00432-f002:**
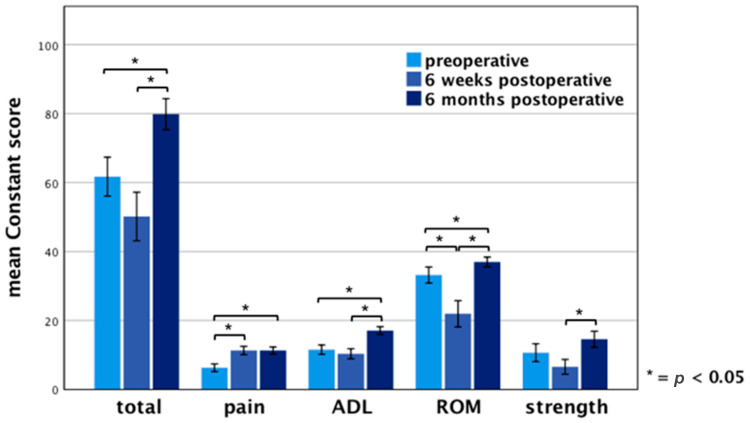
Results of the Constant score on the operated side. Error bars indicate 95% confidence interval (CI). Significant differences (* *p <* 0.05) are designated by asterisks. ADL: activities of daily living ROM: active range of motion.

**Figure 3 jcm-10-00432-f003:**
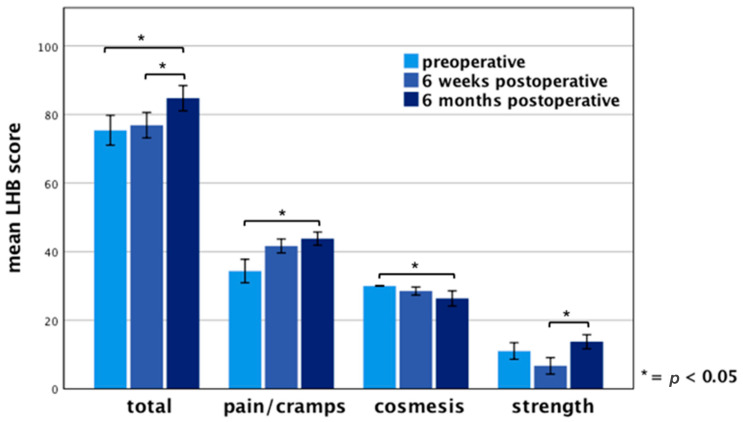
Results of LHB score on the operated side. Error bars indicate 95% CI. Significant differences (* *p <* 0.05) are designated by asterisks.

**Figure 4 jcm-10-00432-f004:**
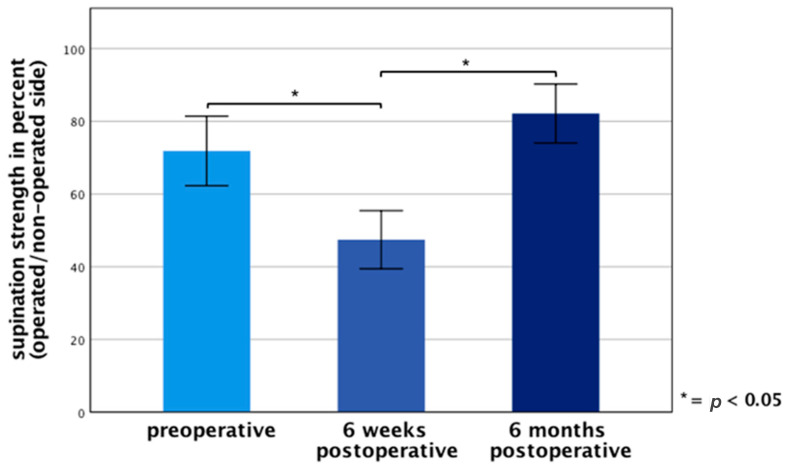
Side-by-side comparison of supination strength in percent. Error bars indicate 95% CI. Significant differences (* *p <* 0.05) are designated by asterisks.

**Table 1 jcm-10-00432-t001:** Demographic data and baseline characteristics of study patients (*n* = 37).

**Sex**		
Female	*n*	11
Male	*n*	26
Age (years)	Mean ± SD	52.0 ± 6.9
BMI (kg/m^2^)	Mean ± SD	28.6 ± 3.8
**Operated side**		
Right shoulder	*n*	23
Left shoulder	*n*	14
Dominant side affected	*n*	24

BMI: body mass index.

**Table 2 jcm-10-00432-t002:** Intraoperative findings of LHB pathologies and concomitant procedures of the study patients (*n* = 37).

Intraoperative Findings of LHB Pathologies		
Partial LHB tear	*n*	37
Tendinitis	*n*	12
Pulley lesion	*n*	35
SLAP lesion	*n*	1
Concomitant surgeries		
Rotator cuff repair	*n*	37
ASAD	*n*	37
ARAC	*n*	1

LHB: long head of biceps SLAP: superior labrum from anterior to posterior ASAD: arthroscopic subacromial decompression ARAC: arthroscopic resection of the acromioclavicular joint.

**Table 3 jcm-10-00432-t003:** Upperarm deformity in study patients (*n* = 37) 6 months postoperative.

Upperarm Deformity		Patient-Dependent	Examiner-Dependent
mild	*n*	2	11
moderate	*n*	2	2
severe	*n*	1	1
total	*n*	5	14

## Data Availability

The data presented in this study are available on request from the corresponding author.

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
