# Peer review of "The Loop Tenodesis Procedure—From Biomechanics to First Clinical Results"

_jcm, 2021, doi:10.3390/jcm10030432_

Round 1

Reviewer 1 Report

In this paper authors This work presented the first short-term clinical results of the novel loop tenodesis procedure for the LHBT disorders. This study is inline with current trends in shoulder arthroscopy procedures and is a specific continuation of authors previous investigations, according to data from the Introduction section.

Overall this is a nice paper – of course it has some limitations – predominantly there is a lack of a proper control group (as admitted by the authors in the limitations). As this is a new technique, this can be understood, yet a substantial review is required before it can be accepted

I have following remarks regarding the paper :

49: You should clearly define the aim of this study for readers, and before the aim, the information about Your previous biomechanical study should be included, because this is a kind of continuum.

51 - 61 – I really have a bit of a problem with reprinting images regarding biomechanical data from other studies. Please consider rewriting this section in a form similar to this : In our previous study we investigated…. briefly, we observed that …. – your paper describing this study is open-access so the readers will have no problems reading it.

70: Inclusion criteria – what about the RCTs, did You classify the lesions?

76: Please, incorporate the Bioethics C. number.

80: Surgical technique – cut by what? scissors, vaporizer? 84 : please provide more details regarding the reinsertion technique – any alterations depending on the extent of the lesion ?

136: Ultrasounds – please incorporate the model of device as well as arthroscope.

141: Did You check, among the failed autotenodesis patients, was the loop below the biceps pulley or somewhere else? Again – the details regarding the surgical technique ….

149: What happened to the 18 subjects, who did not undergo the final follow up? Please, explain.

149 – 155 – this table / description should be placed in the materials section – you merely provide demographic data

158: LHBT tendinitis – how did You recognize the disorder? Did some of them had simultaneous partial tear? Did the abundant vascularization of the tendon structure suggest You the inflammation ? Please touch this subject in the discussion – the diagnosis is not always that obvious - see “Is the inflammation process absolutely absent in tendinopathy of the long head of the biceps tendon? Histopathologic study of the long head of the biceps tendon after arthroscopic treatment.” DOI: 10.5114/pjp.2017.73928.

Authors clearly presented, that the inflammatory cells are occasionally seen in the chronic tendinopathy of biceps, just as the present study subjects.

160: SLAP- which type?

160: RCTs- massive, partial, single tendon, SSC, SST, IST? Please be precise. On the other hand, 35 patients had a biceps pulley lesion and I conclude that SSC and SST injury was common. And pulley lesions – were there any differences between the lesions – did it influence the technique – ex. If the pulley was “almost OK” would you have done the same as in cases where the pulley was “worse than death”

205: According to differences in patients/surgeon observations of the arm cosmetic deformity, it will be valuable if you compare the data of clinical scores in these groups. E.g. compare the patients with Popeye def. and without, in patients – dependent and surgeon – dependent groups, using the statistical test. Also please see for instance this paper

224: What about this patient, where was the loop localized? Was it hyper/hypoechoic or maybe it was completely absent?

295: In the Discussion Section –  you only had one popeye deformity, yet a substantial portion of patients had any/some/mild deformity – in a different study  (“The surgical treatment of the long head of biceps tendon and the autotenodesis phenomenon: an ultrasound and arthroscopic study”; 10.5603/FM.a2019.0072),  with a relatively long follow-up and massive RCTs The Popeye deformity in 15 individuals; however, no patient complained on the visual appearance of arm contour. Also this paper included a sonographic verification of the tenodseiss

302: The limitations, there is also no arm diameter measurements in the Methods section, what can clearly show if there is a Popeye deformity.

Reviewer 2 Report

Line 161 "In addition to the loop tenodesis, all patients received a rotator cuff repair as well as an arthroscopic subacromial decompression. "

This reviewer wonders what the exact degree of the degenerative rotator cuff tears in all subjects was. Was this group homogenic? If so, what was the extent of the cuff pathology? As discussed in Lines 261 and below, functional results are determined not only by LHB procedure performed but also by effectiveness of the RCT treatment. 

Line 115, could you please provide more details on the use of " an upper-arm bandage"? Could this procedure impact on the results? Please discuss "alterations of the postoperative rehabilitation protocol" (line 116) that took place in your study in relation to latest publication of Zabrzynski, Huri et al. (https://doi.org/10.3390/jcm9123938).

Some small suggestions:

Line 76, misspelling; "surgent" should be "surgeon".

Figure 2 legend; "close to its base" -> "insertion" seems to be more appropriate. "Caput humeri" -> "humeral head", English is currently used in the literature instead of Latin.

Round 2

Reviewer 1 Report

It was a pleasure reading the reviewed paper, thank you for sending me this paper for review.